# SEPT: Towards Efficient Scene Representation Learning for Motion Prediction

**Zhiqian Lan**[*], **Yuxuan Jiang**[*], **Yao Mu, Chen Chen, Shengbo Eben Li**[†]
School of Vehicle and Mobility, Tsinghua University
{lanzq21, jyx21}@mails.tsinghua.edu.cn
lishbo@tsinghua.edu.cn

## Abstract

Motion prediction is crucial for autonomous vehicles to operate safely in complex traffic environments. Extracting effective spatiotemporal relationships among traffic elements is key to accurate forecasting. Inspired by the successful practice of pretrained large language models, this paper presents SEPT, a modeling framework that leverages self-supervised learning to develop powerful spatiotemporal understanding for complex traffic scenes. Specifically, our approach involves three masking-reconstruction modeling tasks on scene inputs including agents' trajectories and road network, pretraining the scene encoder to capture kinematics within trajectory, spatial structure of road network, and interactions among roads and agents. The pretrained encoder is then finetuned on the downstream forecasting task. Extensive experiments demonstrate that SEPT, without elaborate architectural design or manual feature engineering, achieves state-of-the-art performance on the Argoverse 1 and Argoverse 2 motion forecasting benchmarks, outperforming previous methods on all main metrics by a large margin.

## 1 Introduction

Accurately predicting the future trajectories of surrounding road users is crucial to a safe and efficient autonomous driving system. In addition to kinematic constraints, the future motions of traffic agents may be shaped by many factors in the traffic scene, including shape and topology of roads, and surrounding agents' behaviors. Modeling and understanding these intricate relationships within the scene has long been a core challenge for motion prediction.

Researches in the early stage predominantly use rasterized semantic images to represent the whole scene from a top-down view, and fuse the information with convolutional neural networks (Lee et al., 2017; Chai et al., 2020; Phan-Minh et al., 2020). Due to the information loss during rasterization, recent researches have shifted to a vectorized paradigm in which the agents and roads are represented as a set of vectors. This representation has served as the foundation for numerous advanced methods that employ graph neural networks and transformers for scene encoding (Liang et al., 2020; Zhou et al., 2022). However, these prevailing SOTA methods typically embrace sophisticated architectural designs, which often rely on anchor-based modeling (Wang et al., 2022) or proposal-refinement scheme (Shi et al., 2022; Wang et al., 2023b) to enhance the prediction performance. These empirical techniques consequently lead to an intricate information processing pipeline.

While previous attempts on scene encoding have focused on innovations in feature engineering and architectural design, we believe that the encoders built on universal architectures can develop strong comprehension on traffic scenes through a properly designed training scheme. A promising direction is to explore self-supervised learning (SSL). Large language models like GPT-3 (Brown et al., 2020) have leveraged SSL on large text corpora to learn broadly applicable linguistic knowledge, achieving significant advancement on a diverse set of NLP tasks. This inspires us that motion prediction models can implicitly be endowed with useful knowledge about traffic scenes such as environment dynamics and social interactions by effective self-supervised objectives.

---

[*]Equal contribution.
[†]Correspondence to Shengbo Eben Li, email: lishbo@tsinghua.edu.cn

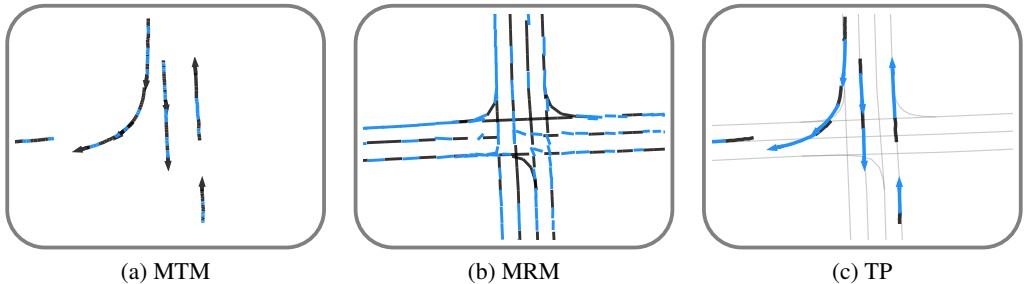

|   (a) MTM   |   (b) MRM   |   (c) TP   |

Figure 1: Illustration for our proposed scene understanding training tasks. For each picture, the masked or truncated scene inputs are painted in black while the corresponding reconstructed results are painted in blue.

Driven by this inspiration, we propose Scene Encoding Predictive Transformer (SEPT), a neat and powerful motion prediction framework that leverages SSL to progressively develop the spatiotemporal understanding for traffic scenes. We construct the self-supervised pretraining scheme for SEPT's scene encoder to capture three key aspects of scene context: temporal dependency within historical trajectory, spatial structure of road network, and interactions among roads and agents, yielding three corresponding pretraining tasks: Masked Trajectory Modeling (MTM), Masked Road Modeling (MRM), and Tail Prediction (TP). MTM, as shown in Figure 1a, randomly masks and reconstructs some waypoints in the agents' trajectories, to encode the temporal dependency arising from kinematic constraints. Similarly, MRM (Figure 1b) randomly masks some portion of the road vectors and then predicts the masked part, allowing the encoder to effectively capture the topology and connectivity of road network. While the previous two tasks handle single input modality, the third task TP (Figure 1c) focuses on interactions between modalities by conducting a short-term motion prediction task. In this task, we divide the agents' trajectories into two sections, named as head and tail, and the objective is to predict the tail section based on the preceding head section and the road context. The pretrained encoder is then finetuned to the downstream motion prediction task. Compared to several recent works exploring SSL in motion prediction (Chen et al., 2023; Cheng et al., 2023), our method adopts a unique pretraining task design and achieves better prediction performance.

The effectiveness of the proposed SEPT are demonstrated by extensive experiments from two aspects. First, the model with pretraining achieves significant improvement over the model learned from scratch on all motion forecasting metrics consistently, showing that the model gains beneficial knowledge through pretraining objectives. Second, the three self-supervised pretraining tasks effectively collaborate with each other and yield positive effects on the final performance in an additive manner. Meanwhile, our experimental results on Argoverse 1 and Argoverse 2 datasets establish SEPT as a top-performing model, ranking 1st across all primary metrics on both large-scale motion forecasting benchmarks. Moreover, compared to the strongest baseline (Zhou et al., 2023) to our knowledge, our model achieves twice faster inference speed with only 40% network parameters.

## 2   RELATED WORKS

**Scene encoding with transformers.** Inspired by the remarkable success of transformer architecture, attention mechanism has been extensively used in motion prediction to model the long-range and complicated interactions within the traffic scene. Early attempts have employed attention in specific sub-modules to encode spatial relationships such as social interactions (Mercat et al., 2020; Salzmann et al., 2020), or agent-map relationships (Huang et al., 2022). A few works model traffic elements as graph and adopt transformer structure to exploit relations in graph-based inputs (Wang et al., 2022; Jia et al., 2023). Recent SOTA approaches, on the other hand, tend to process all input modalities, including trajectories and road network, jointly with hierarchically stacked transformer blocks (Ngiam et al., 2022; Nayakanti et al., 2023; Zhou et al., 2023). These approaches represent the whole traffic scene as 3-D tensors (space-time-feature) and perform factorized attention alternately along the temporal and spatial axes for several rounds to exploit spatiotemporal dependencies. Though sharing the same architectural design concept, the proposed SEPT adopts a more compact

information processing pipeline, where temporal and spatial information are encoded sequentially. This leads to a neat model architecture with fewer functional blocks.

**Self-supervised learning in motion prediction.** Some previous studies have explored self-supervised learning in motion prediction. To the best of our knowledge, the works most closely related to our research are Traj-MAE (Chen et al., 2023) and Forecast-MAE (Cheng et al., 2023). Following the idea of masked autoencoder (He et al., 2022), Traj-MAE designs two independent mask-reconstruction tasks on trajectories and road map input, to train its trajectory and map encoder separately. Such separation leads to a notable limitation that the spatial relationship between agents and roads are not stressed during pretraining. In contrast, SEPT introduces the Tail prediction (TP) pretraining task, which considers both trajectories and roads inputs to capture the spatiotemporal relationships in the traffic scene. Our ablation study has shown the significant impact of TP on the final performance. The other work, Forecast-MAE, adopts a distinct masking strategy for agent trajectory. This approach incorporates the ground truth future trajectories of agents into its pretraining stage and predict the future given its history or vice versa. Its main difference with our SEPT is the granularity of the tokenization for the scene inputs: it takes the whole historical trajectory or a polyline of lane segments as a single token. SEPT, on the other hand, treats a waypoint in a trajectory or a short road vector as a token, which can better capture the motion patterns, as well as the dependency between motions and road structure. In addition, not all self-supervisory tasks in Forecast-MAE contribute to the downstream prediction task positively. As is reported in its ablation study, several combinations of tasks may degrade the final performance compared to models learned from scratch. Instead, SEPT's ablation studies show that all task combinations could improve performance in a consistent and additive manner.

## 3 APPROACH

### 3.1 INPUT REPRESENTATION

SEPT represents the traffic scene input with the following modalities:

**Agent trajectories** are represented as a tensor $[A, T, D_h]$, which captures the recent trajectories of $A$ nearest traffic agents relative to the target agent, including itself. Each agent's trajectory is represented as a time series of $T$ state vectors, containing coordinates, timestamp, agent type, and any other attributes provided in the dataset. All position coordinates are transformed into the local coordinate system of the target agent at its $T$-th frame. In addition, not all agents have full $T$ frames of history as the target agent. Shorter agent history is padded to $T$ frames and properly masked for further attention computation.

**Road network** is represented as a set of $R$ road vectors $[R, D_r]$, based on the vectorized map representation from VectorNet (Gao et al., 2020). Each road vector is a directed lane centerline segment with features including start and end positions, length, turn direction, and other semantics from the dataset. To obtain fine-grained inputs for the road network, lane centerlines are segmented into vectors spanning no longer than 5 meters in length. Representing the entire road network in this way results in an input vector set of massive scale, bringing excessive computational and memory burden in self-attention modules. SEPT introduces a *road pruning module* based on the idea of static path planning (Guan et al., 2022; Jiang et al., 2023), to identify potential routes feasible for traversal by the target agent within the predictive horizon and construct a more compact subset $[S, D_r]$.

### 3.2 MODEL ARCHITECTURE

Figure 2 illustrates the simple but expressive model architecture of the proposed SEPT. The model comprises an encoder for scene encoding and a decoder that predicts a weighted set of trajectories based on scene memory embeddings.

**Projection layer** projects different input modalities into a shared high dimensional vector space $\mathbb{R}^D$ in order to perform follow-up attention operations. A projection layer in SEPT is a single linear layer with ReLU activation, i.e., $\text{Project}(x) = \max(Wx + b, 0)$.

**TempoNet** consists of $K_T$ stacked Transformer encoder blocks, and is used for agent history encoding. TempoNet takes as input agent history embeddings $[A, T, D]$ and conducts self-attention

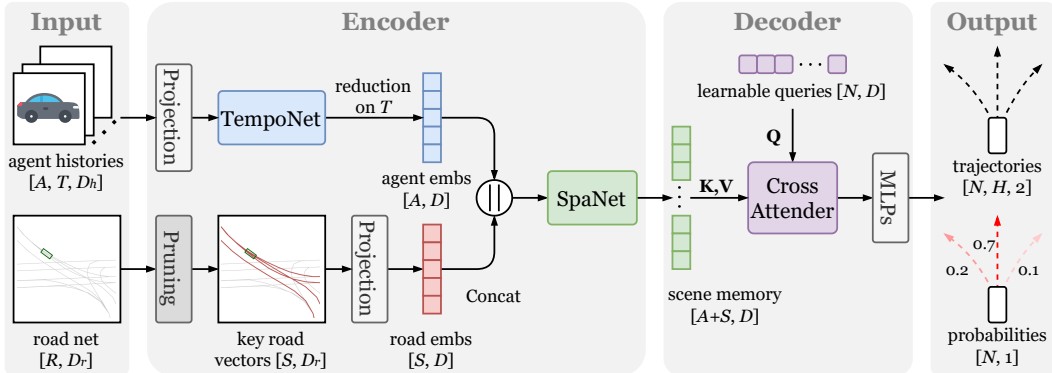

Figure 2: The overall architecture of SEPT

operation along temporal dimension $T$. After that, the embeddings across time steps are aggregated through max pooling to produce agent embeddings $[A, D]$. Additionally, TempoNet applies the simplified relative position embedding proposed in T5 (Raffel et al., 2020), to encode relative order of time steps.

**SpaNet** concatenates $K_S$ Transformer encoder blocks as TempoNet does, and its primary objective is to capture spatial relationships within the traffic scene. Its scene embedding input is formed by concatenating agent and road embeddings. Notably, compared to many previous SOTA methods, SEPT avoids the use of dedicated modules that separately encode agent interaction, road network, and agent-road relations. Instead, SpaNet leverage self attention across spatial dimension $A + S$ to fulfill these encoding objectives in a unified way.

**Cross Attender** comprises $K_C$ stacked cross attention layers. This module cross attends a set of $N$ learnable queries $[N, D]$ with the scene memory embeddings $[A + S, D]$ from the encoder. This generates $N$ embeddings $[N, D]$ corresponding to the $N$ output modalities. Each embedding vector is then mapped from the hidden space to the physical world through two MLPs, producing the predicted 2D trajectory and its associated confidence score.

### 3.3 SCENE UNDERSTANDING TRAINING

In the scene understanding training stage, we train the encoder of SEPT to understand the temporal and spatial relationships in the traffic scene, and produce high quality latent representations for the downstream trajectory prediction task. This is achieved with three self-supervised mask-reconstruction learning tasks on TempoNet, SpaNet, and their alignment, as is shown in Figure 3. In this stage, the model is optimized with the summation of the three tasks' objectives.

#### 3.3.1 TASK 1: MASKED TRAJECTORY MODELING (MTM)

MTM targets TempoNet for understanding the temporal dependencies in agents' trajectories. As is shown in Figure 3a, we randomly substitute a certain ratio of frames with a learnable mask token for all eligible input trajectories. By eligible here, we refer to trajectories longer than a certain threshold, because too short trajectories are often of low quality, and may not be informative enough for the model to learn meaningful representations. For MTM and two more scene understanding tasks below, the hidden vectors are fed into a shallow MLP decoder to produce reconstruction vectors, and the objective function is defined as the masked $L_2$ loss between reconstruction and original vectors.

#### 3.3.2 TASK 2: MASKED ROAD MODELING (MRM)

As is shown in Figure 3b, MRM shares a similar idea with MTM but it targets SpaNet to learn the spatial structure of road vector input. Different from the trajectories in MTM, road segments form a graph rather than a sequence, where simple positional encoding becomes unsuitable. Without bias about relative position information, learning with the token-level mask would be difficult, since a

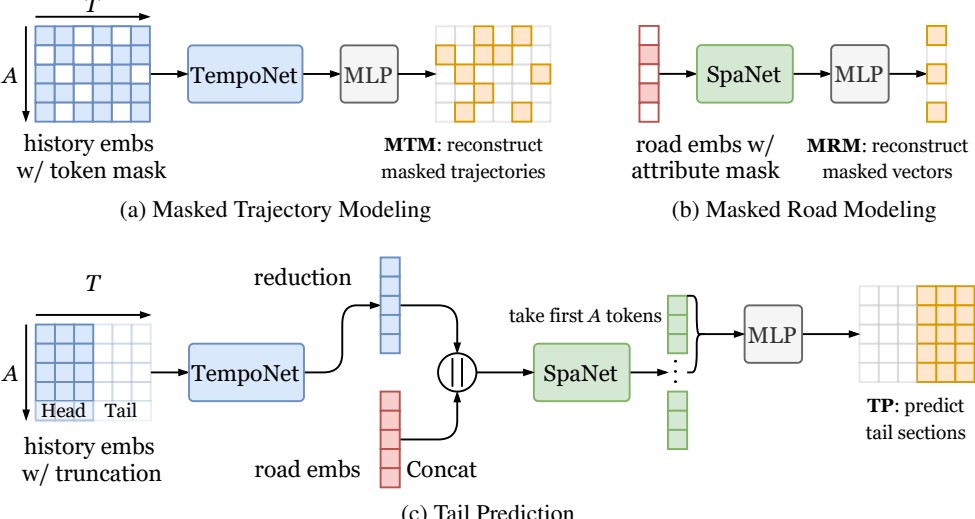

Figure 3: SEPT tasks to learn the temporal and spatial relationship in the traffic scene

time consuming linear assignment problem needs to be solved to compute the reconstruction loss. We instead employ an attribute-level mask technique by setting all attributes, except for the coordinates of the starting point, to zero for selected road feature vectors. The starting point serves as a hint for SpaNet to reconstruct the original vectors. In doing so, SpaNet has to be aware of the connectivity and continuity among unordered road vectors in order to recover the masked attributes like coordinates of endpoint.

### 3.3.3   TASK 3: TAIL PREDICTION (TP)

In TP, the agents' historical trajectories are divided into two sections along the trajectory axis: the head and the tail. The model is trained to predict the tail section given the head section by employing TempoNet and SpaNet jointly, which can be seen as a simplified version of motion prediction, as is shown in Figure 3c. Specifically, the agents' historical trajectories are truncated to the first $T_h$ tokens before fed into TempoNet. Trajectories shorter than a threshold $kT_h$ are excluded from prediction. Then, the embedding vectors of the truncated trajectories and the road embeddings are concatenated and fed into SpaNet. Finally, the hidden vectors of corresponding agents are fed into a shallow MLP decoder to predict the tail sections. This task is designed to align the trajectory representation learned in MTM with the road context representation learned in MRM and model their relationships, since accurate prediction requires effective use of both spatial and temporal information.

### 3.4   MOTION PREDICTION TRAINING

In the motion prediction training stage, we concatenate the pretrained encoder with the decoder, and train the whole model on the labeled dataset in an end-to-end fashion. For the motion prediction task, the loss function $\mathcal{L}$ consists of two terms: the trajectory regression loss $\mathcal{L}_{reg}$ and the classification loss $\mathcal{L}_{cls}$, which are defined in (1).

$$\mathcal{L} = \mathcal{L}_{reg} + \mathcal{L}_{cls},$$
$$\mathcal{L}_{reg} = L_1 Loss(\tau_j, \tau_{gt}), \qquad (1)$$
$$\mathcal{L}_{cls} = -\log p_j,$$

where $\tau_i$, $p_i$ for $i = 1, \ldots, N$ are predicted trajectories with their normalized probabilities, and $j$ refers to the index of the trajectory closest to ground truth in terms of average displacement error (ADE):

$$j = \arg\min_i ADE(\tau_i, \tau_{gt}).$$

## 4 EXPERIMENTS

### 4.1 EXPERIMENTAL SETTINGS

**Dataset.** The effectiveness of our approach is verified on Argoverse 1 and Argoverse 2, two widely-used large-scale motion forecasting datasets collected from real world. The Argoverse 1 dataset consists of $324\,557$ driving scenarios, with each scenario contains 2-second historical context and 3-second future to predict. In contrast, the Argoverse 2 dataset comprises $250\,000$ scenarios, characterized by an extended 5-second historical context and a longer prediction horizon of 6 seconds. The trajectories in both datasets are sampled at $10\,\text{Hz}$.

**Metrics.** Following the evaluation protocol used by Argoverse competition, we calculate the following metrics:

- **minFDE$_k$:** the $l_2$ distance between the endpoint of the best of $k$ forecasted trajectories and the ground truth.
- **minADE$_k$:** the average $l_2$ distance between the best of $k$ forecasted trajectories and the ground truth.
- **miss rate (MR$_k$):** the ratio of scenarios where minFDE$_k$ exceeds a threshold of $2\,\text{m}$.
- **b-minFDE$_k$:** the minFDE$_k$ added by $(1.0 - p)^2$ where $p$ is the probability of the best forecasted trajectory.

The best of $k$ trajectories refers to the trajectory with its endpoint closest to ground truth endpoint. These metrics are calculated for $k = 6$ and $k = 1$, except for b-minFDE$_k$, which is only calculated for $k = 6$.

**Training.** In the scene understanding training stage, we concatenate train, validation and test dataset as the pretrain dataset with labels dropped. The model is trained for 150 epochs with a constant learning rate of $2 \times 10^{-4}$. In the downstream motion prediction training stage, we train and validate following the split of the Argoverse dataset. The model is trained for 50 epochs with the learning rate decayed linearly from $2 \times 10^{-4}$ to 0. Both stages are trained with a batch size of 96 on a single NVIDIA GeForce RTX 3090 Ti GPU.

### 4.2 EXPERIMENTAL RESULTS

**Comparison with baselines.** We compare SEPT performance with SOTA published methods on the Argoverse 1 and Argoverse 2 leaderboards, including DenseTNT Gu et al. (2021), HOME Gilles et al. (2021), MultiPath++ Varadarajan et al. (2022), GANet Wang et al. (2023a), MacFormer Feng et al. (2023), DCMS Ye et al. (2023), Gnet Gao et al. (2023), Wayformer Nayakanti et al. (2023), ProphNet Wang et al. (2023b) and QCNet Zhou et al. (2023) , as is shown in Table 1 & 2.

| Method | **b-minFDE$_6$** | minADE$_6$ | minFDE$_6$ | MR$_6$ | minADE$_1$ | minFDE$_1$ | MR$_1$ |
|---|---|---|---|---|---|---|---|
| DenseTNT | 1.976 | 0.882 | 1.281 | 0.126 | 1.679 | 3.632 | 0.584 |
| HOME | 1.860 | 0.890 | 1.292 | **0.085** | 1.699 | 3.681 | 0.572 |
| MultiPath++ | 1.793 | 0.790 | 1.214 | 0.132 | 1.624 | 3.614 | 0.565 |
| GANet | 1.790 | 0.806 | 1.161 | 0.118 | 1.592 | 3.455 | 0.550 |
| MacFormer | 1.767 | 0.812 | 1.214 | 0.127 | 1.656 | 3.608 | 0.560 |
| DCMS | 1.756 | 0.766 | 1.135 | 0.109 | 1.477 | 3.251 | 0.532 |
| Gnet | 1.751 | 0.789 | 1.160 | 0.117 | 1.569 | 3.407 | 0.545 |
| Wayformer | 1.741 | 0.768 | 1.162 | 0.119 | 1.636 | 3.656 | 0.572 |
| ProphNet | 1.694 | 0.762 | 1.134 | 0.110 | 1.491 | 3.263 | 0.526 |
| QCNet | 1.693 | 0.734 | 1.067 | 0.106 | 1.523 | 3.342 | 0.526 |
| **SEPT (Ours)** | **1.682** | **0.728** | **1.057** | 0.103 | **1.441** | **3.178** | **0.515** |

Table 1: Argoverse 1 leaderboard results sorted by b-minFDE$_6$. The best entry for a metric is marked bold, and the second best is underlined.

| Method | b-minFDE$_6$ | minADE$_6$ | minFDE$_6$ | MR$_6$ | minADE$_1$ | minFDE$_1$ | MR$_1$ |
|---|---|---|---|---|---|---|---|
| MacFormer | 1.905 | 0.701 | 1.377 | 0.186 | 1.838 | 4.686 | 0.612 |
| Gnet | 1.896 | 0.694 | 1.337 | 0.180 | 1.724 | 4.398 | 0.588 |
| ProphNet | 1.882 | 0.657 | 1.317 | 0.179 | 1.764 | 4.768 | 0.610 |
| QCNet | _1.779_ | _0.619_ | _1.191_ | _0.144_ | _1.563_ | _3.962_ | _0.548_ |
| **SEPT (Ours)** | **1.736** | **0.605** | **1.151** | **0.137** | **1.485** | **3.700** | **0.545** |

Table 2: Argoverse 2 leaderboard results.

On the Argoverse 1 dataset, our result outperforms all previous entries and ranks 1st on all metrics except for MR$_6$. For MR$_6$, HOME employs a "sparse sampling" technique to specialize miss rate performance while sacrifices other metrics. In this case, SEPT, being a general purpose method, ranks 2nd. Furthermore, for the Argoverse 2 dataset, SEPT consistently maintains its leading position on the leaderboard, ranking 1st on all metrics. Moreover, compared to QCNet, the strongest baseline in both benchmarks, SEPT has only 40% of its network parameters (around 9.6 million) while exhibits inference speed which is twice faster.

**Comparison with related works.** As is discussed in related works, Traj-MAE and Forecast-MAE share a similar idea of mask-reconstruction for trajectory prediction, but struggle to reach SOTA performance to justify pretraining. Traj-MAE reports minADE$_6$, minFDE$_6$ and MR$_6$ on Argoverse 1 test set, and Forecast-MAE reports these metrics on Argoverse 2 test set. As is shown in Table 3, SEPT outperforms these two methods by a large margin, further demonstrating our method's effectiveness.

| Method | minADE$_6$ | minFDE$_6$ | MR$_6$ |
|---|---|---|---|
| Traj-MAE | 0.81 | 1.25 | 0.137 |
| SEPT | **0.728** | **1.057** | **0.103** |

| Method | minADE$_6$ | minFDE$_6$ | MR$_6$ |
|---|---|---|---|
| Forecast-MAE | 0.690 | 1.338 | 0.173 |
| SEPT | **0.605** | **1.151** | **0.137** |

(a) Comparison with Traj-MAE on Argoverse 1

(b) Comparison with Forecast-MAE on Argoverse 2

Table 3: Performance comparison with related works

**Trajectory prediction visualization.** We compare the prediction results between the model with pretraining and the model trained from scratch in Figure 4. The visualization demonstrates that the model with pretraining can better capture the multimodality of driving purposes (shown in case 1) and generate the trajectories with improved conformity to road shapes (shown in case 2 & 3).

### 4.3 ABLATION STUDIES

The ablation studies focus on the scene understanding stage, specifically the effectiveness of various combinations of scene understanding tasks and the impact of their hyperparameters. We conduct ablation experiments following the same routine as the main experiment, and report the evaluation performance on the Argoverse 1 validation dataset. To minimize the impact of randomness, we conduct multiple experiments with different seeds for each configuration.

**Effectiveness of scene understanding tasks.** MTM, MRM and TP are three tasks introduced to learn a good prior for the downstream trajectory prediction task. To investigate the effectiveness, we conduct experiments for all 8 combinations, and the results are shown in Figure 5 and Table 4.

In Figure 5, we use solid lines for combinations involving task TP, and dashed lines for combinations without this task. It is clear that combinations with TP consistently outperform the opposites by learning faster at the beginning and performing better after convergence. This indicates that TP is effective to align TempoNet and SpaNet, therefore improve the overall representation quality of scene encoding. The other two tasks, MTM and MRM, can also improve performance in a consistent and additive manner. Furthermore, it should be noted that the configuration with all tasks active not only achieves the best performance, but also has the lowest variance over 5 runs among all combinations, as is shown in the shaded area in Figure 5. This suggests that our proposed training approach can help the model converge to a good level consistently. Table 4 includes the

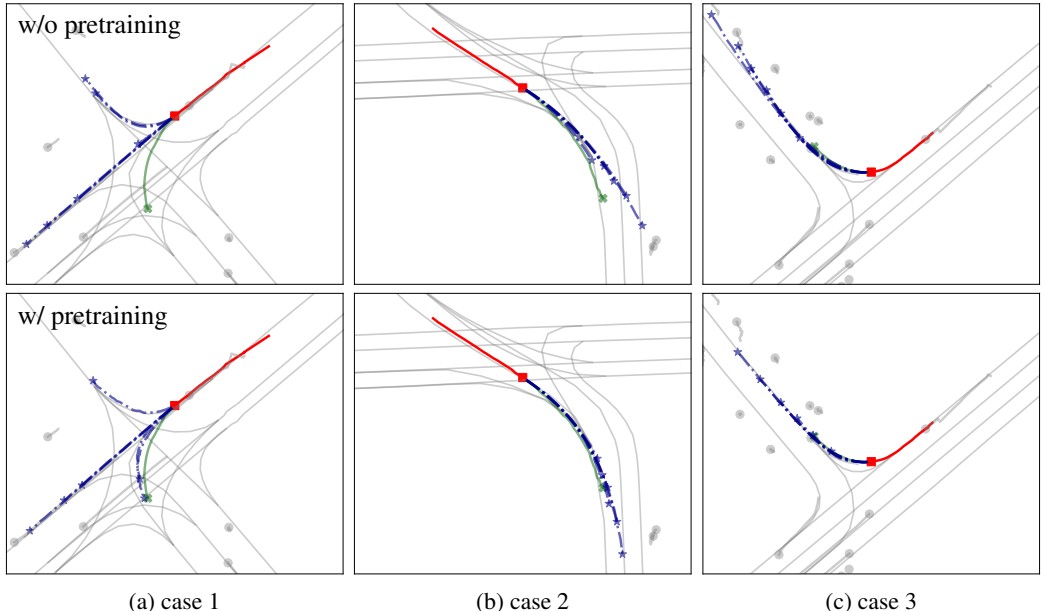

| (a) case 1 | (b) case 2 | (c) case 3 |

Figure 4: Visualization results on three selected scenarios from Argoverse 1 validation set. The target agent's history and ground truth future are shown in red and green, respectively, while model's predictions are shown in blue. In each case, the upper image corresponds to the model that learns from scratch while the lower image corresponds to the model with scene understanding training.

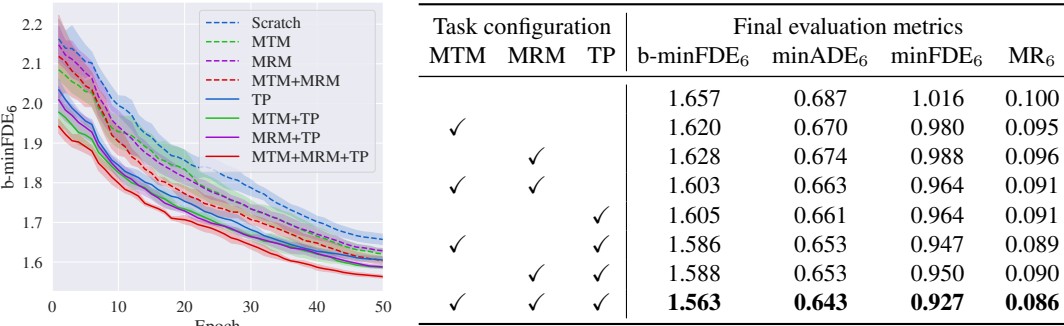

| Task configuration | | | Final evaluation metrics | | | |
|---|---|---|---|---|---|---|
| MTM | MRM | TP | b-minFDE$_6$ | minADE$_6$ | minFDE$_6$ | MR$_6$ |
| | | | 1.657 | 0.687 | 1.016 | 0.100 |
| ✓ | | | 1.620 | 0.670 | 0.980 | 0.095 |
| | ✓ | | 1.628 | 0.674 | 0.988 | 0.096 |
| ✓ | ✓ | | 1.603 | 0.663 | 0.964 | 0.091 |
| | | ✓ | 1.605 | 0.661 | 0.964 | 0.091 |
| ✓ | | ✓ | 1.586 | 0.653 | 0.947 | 0.089 |
| | ✓ | ✓ | 1.588 | 0.653 | 0.950 | 0.090 |
| ✓ | ✓ | ✓ | **1.563** | **0.643** | **0.927** | **0.086** |

Figure 5 & Table 4: The training curves report b-minFDE$_6$ performance for all task combinations. The solid or dashed lines correspond to the mean and the shaded regions correspond to 1-$\sigma$ confidence interval over 5 runs. The table reports the final performance of all $k = 6$ metrics. The best entry for a metric is marked bold.

final performance for all $k = 6$ metrics. It can be seen that other metrics exhibit similar patterns of improvement to b-minFDE$_6$ discussed above. Training curves for those metrics are included in the Appendix.

**Effects of mask hyperparameters.** In this ablation experiment, we show the impact of mask hyperparameters on final performance. For MTM and MRM, the main mask hyperparameter is the mask ratio $p$, implemented as sampling a mask from Bernoulli($p$). For TP, the mask hyperparameter is the length of visible history $T_h$. For each of these three hyperparameters, we pretrain with the single corresponding task, then finetune with identical setting to get the final performance. As is shown in Figure 6, mask hyperparameters do not have a significant impact in suitable ranges and always outperform the model that trains from scratch. This suggests the effectiveness of our task design and

its robustness to choices of hyperparameters. In our main experiment, we simply use $p_{\text{MTM}} = 0.5$, $p_{\text{MRM}} = 0.5$ and $T_h = 8/20$ (for Argoverse 1/2) without tuning.

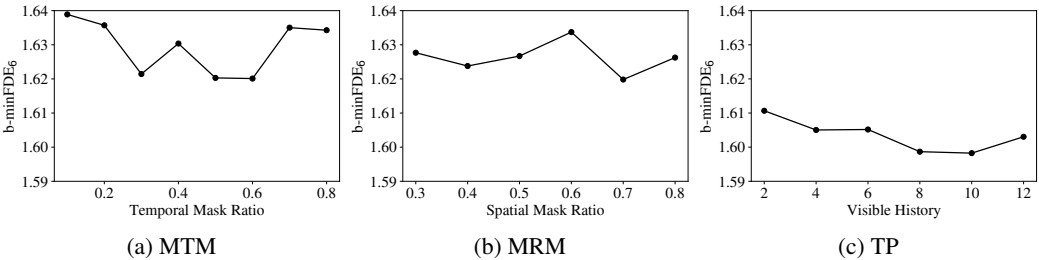

(a) MTM                    (b) MRM                    (c) TP

Figure 6: Final b-minFDE$_6$ performance of varying mask settings for each task, averaged over 5 runs.

## 5 CONCLUSION

In this paper, we present Scene Encoding Predictive Transformer (SEPT), a powerful modeling framework for accurate and efficient motion prediction. Leveraging three self-supervised mask-reconstruction learning tasks for scene understanding, we achieve SOTA performance on two large-scale motion forecasting datasets with a neat and unified model architecture. The ablation studies further demonstrate that introduced pretraining tasks can significantly improve prediction performance in a consistent and additive manner. Meanwhile, we acknowledge the limitation that, due to its agent-centric scene representation, SEPT is a single-agent prediction framework, and is not trivially extendable to multi-agent joint prediction. In the future, we will investigate further and generalize it to a more powerful multi-agent prediction framework.

## 6 ACKNOWLEDGEMENT

This work is supported by NSF China under 52221005. It is also partially supported by Tsinghua University-Didi Joint Research Center for Future Mobility.

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

# 7 APPENDIX

## 7.1 RESULTS OF ABLATION STUDIES

Results of ablation studies for more evaluation metrics are shown in the following figures, which further corroborated the conclusions drawn in the main text.

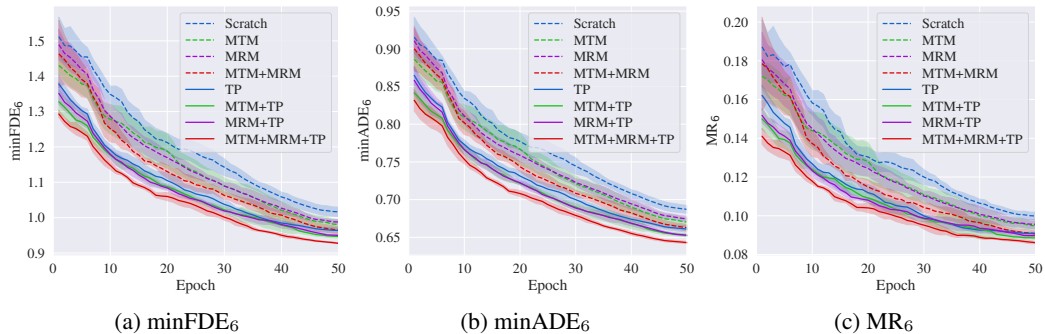

(a) minFDE$_6$      (b) minADE$_6$      (c) MR$_6$

Figure 7: The training curves report minFDE$_6$, minADE$_6$ and MR$_6$ performance of all task combinations. The solid or dashed lines correspond to the mean and the shaded regions correspond to 1-sigma confidence interval over 5 runs.

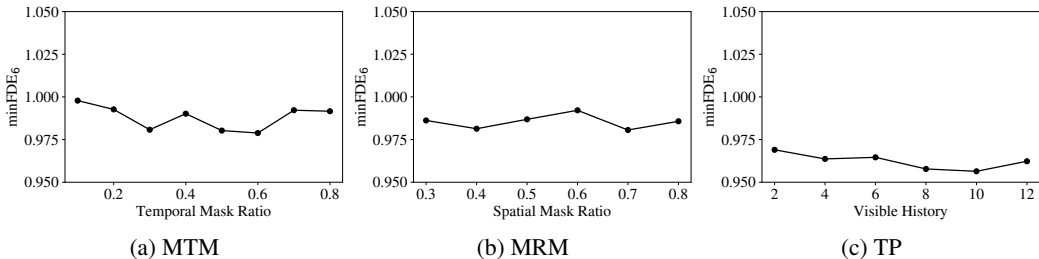

(a) MTM      (b) MRM      (c) TP

Figure 8: Final minFDE$_6$ performance of varying mask settings for each task, averaged over 5 runs.

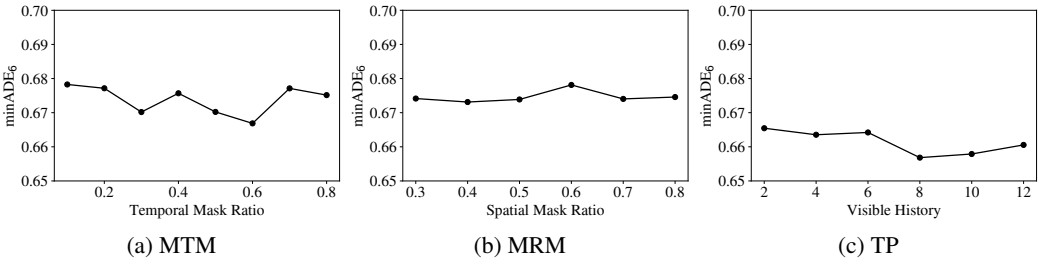

(a) MTM      (b) MRM      (c) TP

Figure 9: Final minADE$_6$ performance of varying mask settings for each task, averaged over 5 runs.

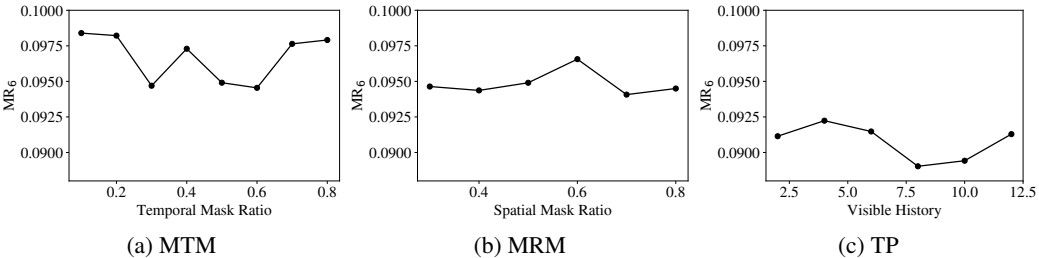

Figure 10: Final $MR_6$ performance of varying mask settings for each task, averaged over 5 runs.

## 7.2 IMPACT OF PRETRAINING DATASET

To assess the impact of pretraining data on the downstream task, we compare the models pretrained on varying datasets: (a) train set, (b) train and validation set, and (c) train, validation and test set (consistent with the main experiment's setting). The performances are evaluated on the validation set, as reported in Figure 11 and Table 5. Our preliminary observations offer two insights. Firstly, there is only minor performance differences between setting (a) and (b). This suggests that the improvement brought about by pretraining cannot be simply attributed to exposure to the dataset for evaluation. Secondly, augmenting pretraining with additional data from the test set also improves performance on validation set. This hints at the notion that pretraining might derive benefits from a more diverse dataset, aligning with the discoveries in LLM pretraining studies.

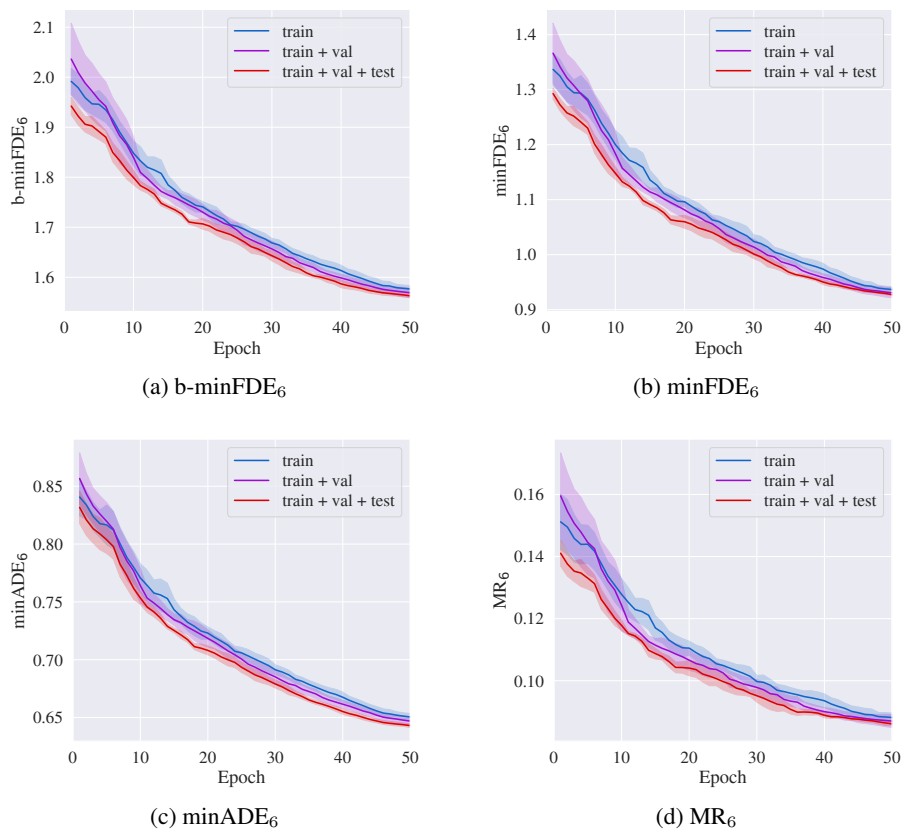

Figure 11: The training curves report b-minFDE$_6$, minFDE$_6$, minADE$_6$ and MR$_6$ performance of models pretrained with different pretraining dataset settings. The solid lines correspond to the mean and the shaded regions correspond to 1-sigma confidence interval over 5 runs.

| Data split | | | Final evaluation metrics | | | |
|---|---|---|---|---|---|---|
| Train | Validation | Test | b-minFDE$_6$ | minADE$_6$ | minFDE$_6$ | MR$_6$ |
| ✓ | | | 1.577 | 0.650 | 0.937 | 0.088 |
| ✓ | ✓ | | 1.569 | 0.647 | 0.931 | 0.087 |
| ✓ | ✓ | ✓ | 1.563 | 0.643 | 0.927 | 0.086 |

Table 5: Final performance of different pretraining dataset settings.

## 7.3 HYPERPARAMETERS

Table 6 reports the hyperparameters for the SEPT network architecture. In all transformer layers, layer norm is done prior to attention and feedforward operations, and bias is cancelled in feedforward networks.

| Arch | Parameters | Values |
|---|---|---|
| Projection | output size | 256 |
| TempoNet | depth | 3 |
| | num_head | 8 |
| | dim_head | 64 |
| SpaNet | depth | 2 |
| | num_head | 8 |
| | dim_head | 64 |
| Cross attender | depth | 3 |
| | num_head | 8 |
| | dim_head | 64 |
| | query size | 256 |
| MLP_traj / MLP_score | num_hidden | 1 |
| | hidden size | 512 |

Table 6: Network hyperparameters

