# OpenReview forum: "SEPT: Towards Efficient Scene Representation Learning for Motion Prediction"
_ICLR.cc/2024/Conference — ICLR 2024 poster_

### Official Review · Reviewer_Ukgp · 2023-10-17

**Soundness:** 3 good
**Presentation:** 2 fair
**Contribution:** 2 fair
**Rating:** 6
**Confidence:** 5

**Summary:**

This paper combines several self-supervised reconstruction objectives as pretraining for motion prediction. The proposed method obtains state-of-the-art performance on two public leaderboards.


Update:
This work demonstrates great performance and their findings might benefit the community, though the structure and idea is not entirely new. I change my score to 6.

**Strengths:**

Strong emperical performance. The proposed method ranks top on two widely used leaderboards.

**Weaknesses:**

- Limited novelty: The model structure is not new, which is basically SceneTransformer [A], which should be considered as contributions. For the three pretraining objectives, it is very similar to Traj-MAE[B] and ForecastMAE[C], as stated by the authors as well. Compared to [B] and [C], they achieve better performance, but the paper does not give a thorough analysis regarding the difference. As a result, it is hard to determine the real working part of the method.

[A]Scene Transformer: A unified architecture for predicting multiple agent trajectories. ICLR 2022

[B]Traj-MAE: Masked Autoencoders for Trajectory Prediction. ICCV 2023

[C]Forecast-MAE: Self-supervised Pre-training for Motion Forecasting with Masked Autoencoders. ICCV 2023

**Questions:**

1. Could the authors explain more about the difference between your method and Traj-MAE/ForecastMAE and discuss why your method better performance than them? It might be the case that "the devil is in the details" and I think it is important to find out these details as your contributions. Otherwise, the proposed methodology is basically the same as Traj-MAE/ForecastMAE, which has nothing new.

2. Do the authors plan to open source the code so that the community could easily reproduce the results and find out the working part?

3. Missed reference: it seems that some classic transformer based works are not mentioned, which might cause wrong understanding of the contributions of this work:

[D] Motion Transformer with Global Intention Localization and Local Movement Refinement. NeurIPS 2022 Oral

[E] HDGT: Heterogeneous Driving Graph Transformer for Multi-Agent Trajectory Prediction via Scene Encoding. IEEE TPAMI

[F] Multi-modal motion prediction with transformer-based neural network for autonomous driving. ICRA 2022


In summary, the proposed method has strong emperical performance on the public leaderboards but the method described in the paper basically has nothing new compared to existing methods, which makes their contributions unclear. Since the authors neither provide any detailed comparisions and analysis with existing methods nor mention their plans for open source, I do not think the paper could provide much value to the community, thus I give a borderline reject.  If the authors could solve the aforementioned issues, I may change my scores accordingly.

---

> ### Author Response · Authors · 2023-11-14
> **Response to Reviewer Ukgp (1/2)**
>
> Thank you for your thorough assessment and valuable feedback on our paper. We have addressed each of your concerns as follows.
>
> > W1 (a): Limited novelty: The model structure is not new, which is basically SceneTransformer [A], which should (not) be considered as contributions.
>
> We consider scene understanding pretraining as the primary contribution of SEPT paper.  Benefited from the pretraining scheme, our model could achieve SOTA performance with a greatly simplified architecture, as illustrated in Figure 2 of the paper. Regarding our model's relationship with SceneTransformer, while both the two models are built on transformer, there are several notable differences in their architectural designs:
> 1. **Encoder architecture**: SceneTransformer takes a spatiotemporal-intertwined encoding structure that performs factorized attention along both the time and space (agent & road graph) axes alternately and repeatedly, to fuse the spatiotemporal information. In contrast, SEPT adopts a spatiotemporal-separated encoding mechanism that, first processes temporal information within agent trajectories and reduces the time dimension by max pooling, then encodes the spatial relationships among agents and roads. This saves computation, allowing for more efficient training and inference.
> 2. **Decoder architecture**: SEPT's decoder uses N learnable queries to aggregate information from scene embeddings through cross-attention operation, with N corresponding to the number of motion modalities. On the other hand, SceneTransformer's decoder conducts self-attention across time and agent dimension on the "agent features" which is stacked N times, and it adds one-hot encoding to distinguish each predictive modality.
>
> > W1 (b): For the three pretraining objectives, it is very similar to Traj-MAE[B] and ForecastMAE[C], as stated by the authors as well. Compared to [B] and [C], they achieve better performance, but the paper does not give a thorough analysis regarding the difference. As a result, it is hard to determine the real working part of the method.
>
> > Q1: Could the authors explain more about the difference between your method and Traj-MAE/ForecastMAE and discuss why your method better performance than them? It might be the case that "the devil is in the details" and I think it is important to find out these details as your contributions. Otherwise, the proposed methodology is basically the same as Traj-MAE/ForecastMAE, which has nothing new.
>
> Regarding the relationship between our method and Traj-MAE and Forecast-MAE, we would like to revise our paper and expand the related works section to highlight the main differences:
>
> 1. **Traj-MAE** designs two independent mask-reconstruction tasks on trajectories and road map input, to train its trajectory and map encoder separately. Such separation leads to a notable limitation that the spatial relationship between agents and roads are not stressed during pretraining. In contrast, SEPT's pretraining introduces the Tail prediction (TP) task, which considers both temporal and spatial information, and jointly trains TempoNet and SpaNet. Our ablation experiments clearly show that TP has a significant impact on the final performance improvement. This could be explained by the fact that TP can help the encoder better align the representations learned by the masked trajectory modeling (MTM) and masked road modeling (MRM) tasks.
> 2. **Forecast-MAE** has a notable difference with our method: the granularity of the tokenization for the scene inputs. Forecast-MAE uses the whole historical or future trajectory as a single token, while SEPT treats a waypoint in a trajectory as a token. Similarly, for road input, it uses a polyline of lane segment as a token, while our method uses a short road vector less than 5m. We believe that our approach can better capture the motion patterns, as well as the dependency between the historical motions and the road structure. Also, Forecast-MAE adopts a distinct masking strategy for agent trajectory by incorporating ground-truth future in pretraining, and it predict the future given its history or vice versa.
>
> Also, we would like to share more thoughts on the performance differences:
> 1. Within the benchmarks of Argoverse 1 and 2, preceding SOTA methods like QCNet and ProphNet set a notable standard. SEPT was able to surpass these methods, but it required the assistance of SSL pretraining to do so. Hence, we consider the proposed SSL pretraining scheme as the primary contributor to the observed performance improvement.
> 2. It's worth mentioning that Traj-MAE chose AutoBots as the baseline model, which, as observed, is not a strong competitor on Argoverse 1 (minADE 0.89; minFDE 1.41). We would like to highlight that earlier this year, our prototype model, without pretraining, achieved a minADE of 0.83 and a minFDE of 1.30 on the Argoverse 1 test set. We believe that the performance improvement over a stronger baseline model lends more credibility to our method.

---

> ### Author Response · Authors · 2023-11-14
> **Response to Reviewer Ukgp (2/2)**
>
> > Do the authors plan to open source the code so that the community could easily reproduce the results and find out the working part?
>
> Certainly! We plan to opensource our codebase once the paper is accepted. It is a great pleasure to share our work with the community!
>
> > Missed reference: it seems that some classic transformer based works are not mentioned, which might cause wrong understanding of the contributions of this work.
>
> We appreciate your suggestion to include more classic transformer-based works in our related works section. We agree that these works are important and relevant to our research topic, and we will review them in our revised manuscript.

---

> > ### Comment · Reviewer_Ukgp · 2023-11-21
> > **Concerns are solved**
> >
> > The technical concerns are solved. I will raise my score. It demonstrates great engineering efforts to modify and make the previous method work.

---

### Official Review · Reviewer_siKy · 2023-10-31

**Soundness:** 3 good
**Presentation:** 4 excellent
**Contribution:** 3 good
**Rating:** 8
**Confidence:** 5

**Summary:**

The paper presents a modeling approaching for the motion prediction problem for AVs that involves pre-training on self-supervised training objectives (masking and prediction of trajectory and road-graph portions) followed by fine-tuning for the downstream motion prediction task.

Pre-training is multi-task, with the masking of intermediate trajectory steps towards capturing motion kinematics, masking of road network vector attributes for better road topology and connectivity understanding, and sequential step masking of trajectory steps to create a motion prediction problem similar to downstream and capture spatio-temporal relationships between agent motion and road network.

This setup and the pre-training tasks with an ablation study of each of their contributions towards the final metrics are the main contributions of the paper. The results are demonstrated on Argoverse 1 and 2 challenges, where the paper achieves SOTA results with a convincing margin.

**Strengths:**

- Pre-training and self-supervised learning for the motion prediction task are under-explored areas, and this paper sets a strong benchmark with strong results on a widely used public leaderboard.

- The authors have opted for a simple stacked-transformer architecture without any bag-of-tricks, that helps keep the focus on the contributions from the pre-training tasks and setup.

- The ablation studies are extensive and well set up, providing insights into contributions from the each of the proposed components.

- Overall presentation of the paper is good and it is easy to follow. Along with the simplistic architecture, it would allow others to reproduce and build upon this work.

**Weaknesses:**

- The paper mentions using the test set (unlabeled wrt the actual task) for self-supervised learning. While the test set from Argoverse *is* available to users, the intention is to provide it so that users can run their own inference and submit the predictions. Using the unlabelled part of this dataset (road network, agent motion histories) for pre-training could be giving the model an unfair advantage. I would encourage the authors to report the results without using the test set anywhere in their training / validation, or at the very least report an ablation result without using the test set for pre-training.

- While results are reported on both Argoverse 1 and 2, it would be good to see generalization of the proposed methods to at-least one other dataset such as waymo open dataset or nuScenes.

**Questions:**

Mainly the one mentioned in the "Weaknesses" section:
- What is the impact on the results when not using the test set for pre-training?

---

> ### Author Response · Authors · 2023-11-14
> **Response to Reviewer siKy**
>
> Thank you for the detailed and insightful review of our paper. We have addressed each of your concerns as follows.
>
> > W1: The paper mentions using the test set (unlabeled wrt the actual task) for self-supervised learning. While the test set from Argoverse is available to users, the intention is to provide it so that users can run their own inference and submit the predictions. Using the unlabelled part of this dataset (road network, agent motion histories) for pre-training could be giving the model an unfair advantage. I would encourage the authors to report the results without using the test set anywhere in their training / validation, or at the very least report an ablation result without using the test set for pre-training.
>
> > Q1: What is the impact on the results when not using the test set for pre-training?
>
> For participants in the Argoverse 1 motion prediction challenge, it's a widely recognized observation that test performance often exhibits a substantial gap when compared to validation performance, potentially due to distribution mismatches. Indeed, our initial motivation for exploring pretraining, especially on the test set, as the reviewer speculates, was to address this gap. However, the experimental results did not align with our expectations. Surprisingly, this led us in a different direction, ultimately resulting in an overall improvement in prediction performance.
>
> During our investigation, we also experimented with excluding the test set from the pretraining process. We found that this exclusion has a negligible impact on the prediction performance. We decide to keep using the test set to demonstrate that our pretraining could incorporate all the data at hand. In the next few days, we intend to conduct further experiments and will include an ablation study to provide additional insight into the use of the test set in our approach.
>
> > W2: While results are reported on both Argoverse 1 and 2, it would be good to see generalization of the proposed methods to at-least one other dataset such as waymo open dataset or nuScenes.
>
> We recognize the importance of testing SEPT's performance on a wider range of datasets to assess its generalization capabilities more comprehensively. Specifically, we will consider including test results on additional datasets like Waymo Open Dataset or nuScenes in the camera-ready version to provide a more thorough evaluation of our approach. Thank you for your feedback, and we will ensure that this aspect is addressed in our final submission.

---

### Official Review · Reviewer_fR6X · 2023-11-01

**Soundness:** 2 fair
**Presentation:** 3 good
**Contribution:** 2 fair
**Rating:** 6
**Confidence:** 4

**Summary:**

The paper presents SEPT (Scene Encoding Predictive Transformer), a framework for motion prediction crucial for autonomous vehicles operating in complex traffic environments. SEPT leverages self-supervised learning to develop a powerful spatiotemporal understanding of complex traffic scenes. The approach involves three masking-reconstruction modeling tasks on scene inputs, including agents’ trajectories and road networks, pretraining the scene encoder to capture kinematics within trajectories, spatial structures of road networks, and interactions among roads and agents. SEPT, without elaborate architectural design or manual feature engineering, achieves state-of-the-art performance on the Argoverse 1 and Argoverse 2 motion forecasting benchmarks, outperforming previous methods on all main metrics by a significant margin. The pretrained encoder is then fine-tuned on the downstream forecasting task, demonstrating that the model gains beneficial knowledge through pretraining objectives.

**Strengths:**

1. SEPT achieves state-of-the-art performance on two large-scale motion forecasting benchmarks (Argoverse 1 and Argoverse 2), outperforming previous methods by a significant margin in terms of various metrics.
2. The ablation studies in the paper show that all combinations of the introduced tasks consistently improve the model’s performance. Each task contributes additively to the enhancement of the model's predictive accuracy.
3. The manuscript is well-written overall and presents its content in a clear manner.

**Weaknesses:**

1. The Introduction effectively illustrates the potential of SSL in motion prediction tasks. However, it could benefit from acknowledging previous works such as Traj-MAE and Forecast-MAE that have already explored SSL in motion prediction. The three self-supervised tasks presented appear to be inspired or derived from these preceding studies.
2. The framework presented in the paper is well-organized and operational, but it seems to lack novelty and groundbreaking contributions in the area of study.
3. It's observed that the predominant improvement in SEPT's performance relative to the previous state-of-the-art isn’t chiefly attributed to SSL pretraining. This seems to be somewhat divergent from the central thesis and focus of the paper.

**Questions:**

1. Regarding the related work, you’ve cited that the lane mask SSL task of Forecast-MAE didn’t exhibit considerable advancement in the ablation study. However, in Traj-MAE, there was a significant enhancement noted in the HDMap mask. Isn't the consistent improvement across all SSL tasks already an established finding prior to this research?
2. In table 4, there is a noted reduction of 6.4% in minADE on the validation set relative to the baseline without SSL pretraining. Meanwhile, SEPT shows a 10.1% and 12.3% decrease in minADE compared to Traj-MAE and Forecast-MAE, respectively. Could you elucidate whether the baseline model, even without the SSL pretraining, has already exceeded the performance of previous state-of-the-art methods? And what are the primary contributors to this enhancement in performance?
3. On the other hand, it seems that in Traj-MAE, its pretraining method improved the baseline's 0.732 minADE to a final 0.604 minADE, marking a relative decrease of 17.5%. Would this imply that Traj-MAE's pretraining technique holds a comparative advantage or superiority over SEPT’s methodology?

---

> ### Author Response · Authors · 2023-11-14
> **Response to Reviewer fR6X (1/3)**
>
> Thank you for the insightful and constructive feedback on our paper. We appreciate the thorough evaluation and your valuable comments, which we have carefully considered and addressed in our response below.
>
> > W1: The Introduction effectively illustrates the potential of SSL in motion prediction tasks. However, it could benefit from acknowledging previous works such as Traj-MAE and Forecast-MAE that have already explored SSL in motion prediction. The three self-supervised tasks presented appear to be inspired or derived from these preceding studies.
>
> We appreciate your feedback and agree that our paper could benefit from acknowledging previous works, including Traj-MAE and Forecast-MAE, which have explored self-supervised learning in motion prediction.
>
> However, we want to clarify that while our work draws inspiration from the success of self-supervised learning, it is not directly derived from Traj-MAE or Forecast-MAE. Early this year, our initial motivation for exploring pretraining was to address the validation-test performance gap of our model on Argoverse 1. The experimental results did not align with our expectations, but led us in a different direction, ultimately resulting in an overall improvement in prediction performance. We became aware of Traj-MAE and Forecast-MAE around June 2023, but by that time, our task design had already been finalized.
>
> We will revise the introduction to include references to these works. Also, to address the concerns raised by multiple reviewers regarding the relationship between our method and Traj-MAE and Forecast-MAE, we would like to expand the related works section to highlight the main differences:
> 1. **Traj-MAE** designs two independent mask-reconstruction tasks on trajectories and road map input, to train its trajectory and map encoder separately. Such separation leads to a notable limitation that the spatial relationship between agents and roads are not stressed during pretraining. In contrast, SEPT's pretraining introduces the Tail prediction (TP) task, which considers both temporal and spatial information, and jointly trains TempoNet and SpaNet. Our ablation experiments clearly show that TP has a significant impact on the final performance improvement. This could be explained by the fact that TP can help the encoder better align the representations learned by the masked trajectory modeling (MTM) and masked road modeling (MRM) tasks.
> 2. **Forecast-MAE** has a notable difference with our method: the granularity of the tokenization for the scene inputs. Forecast-MAE uses the whole historical or future trajectory as a single token, while SEPT treats a waypoint in a trajectory as a token. Similarly, for road input, it uses a polyline of lane segment as a token, while our method uses a short road vector less than 5m. We believe that our approach can better capture the motion patterns, as well as the dependency between the historical motions and the road structure. Also, Forecast-MAE adopts a distinct masking strategy for agent trajectory by incorporating ground-truth future in pretraining, and it predict the future given its history or vice versa.
>
> > W2: The framework presented in the paper is well-organized and operational, but it seems to lack novelty and groundbreaking contributions in the area of study.
>
> Thank you for acknowledging the organization and operational nature of our framework. While we understand that our work may not be deemed groundbreaking in the sense of a revolutionary breakthrough, we do believe that novelty lies in our introduction of self-supervised learning (SSL) to the field of motion prediction, as well as our unique task design. Our experiments and ablation studies have shown that pretraining for scene understanding significantly contributes to prediction performance.
>
> In response to your valuable feedback, we will also revise the paper to provide more in-depth details about the relationship between our method and Traj-MAE as well as Forecast-MAE. This will help clarify our specific contributions in the context of existing research.

---

> ### Author Response · Authors · 2023-11-14
> **Response to Reviewer fR6X (2/3)**
>
> > W3: It's observed that the predominant improvement in SEPT's performance relative to the previous state-of-the-art isn’t chiefly attributed to SSL pretraining. This seems to be somewhat divergent from the central thesis and focus of the paper.
>
> > Q2: In table 4, there is a noted reduction of 6.4% in minADE on the validation set relative to the baseline without SSL pretraining. Meanwhile, SEPT shows a 10.1% and 12.3% decrease in minADE compared to Traj-MAE and Forecast-MAE, respectively. Could you elucidate whether the baseline model, even without the SSL pretraining, has already exceeded the performance of previous state-of-the-art methods? And what are the primary contributors to this enhancement in performance?
>
> Defining 'state-of-the-art' in this context might introduce ambiguity. In our interpretation, SOTA references the leading techniques showcased on specific leaderboards, such as QCNet and ProphNet on Argoverse, as well as MTR on WOMD. Traj-MAE and Forecast-MAE, while exploring SSL, did not attain SOTA performance on both Argoverse 1 and 2.
>
> Within the benchmarks of Argoverse 1 and 2, preceding SOTA methods like QCNet and ProphNet set a notable standard. SEPT was able to surpass these methods, but it required the assistance of SSL pretraining to do so. Hence, we consider the proposed SSL pretraining scheme as the primary contributor to the observed performance improvement.
>
> Regarding the comparison with Traj-MAE and Forecast-MAE, besides pretraining, our performance gains likely stem from various other elements, encompassing data preprocessing, input representation, model architecture, and hyperparameters. Through our exploration, we found that the agent-centric data representation and the DETR-like decoder architecture may have pivotal roles in this observed enhancement.
>
> > Q1: Regarding the related work, you’ve cited that the lane mask SSL task of Forecast-MAE didn’t exhibit considerable advancement in the ablation study. However, in Traj-MAE, there was a significant enhancement noted in the HDMap mask. Isn't the consistent improvement across all SSL tasks already an established finding prior to this research?
>
> We noticed that Traj-MAE indeed conducted an ablation study on the effectiveness of pretraining tasks, as indicated in Table 4 of the paper and Table 4 of the supplementary material. From our perspective, while the consistent improvement across SSL tasks can be considered an indicator of reasonable task design, it may not be the sole criterion for evaluating a method's efficacy.
>
> It's worth mentioning that Traj-MAE chose AutoBots as the baseline model, which, as observed, is not a strong competitor on the Argoverse 1 leaderboard (minADE 0.89; minFDE 1.41). We would like to highlight that earlier this year, our prototype model, without extensive pretraining or hyperparameter tuning, achieved a minADE of 0.83 and a minFDE of 1.30 on the Argoverse 1 test set, with only 3.5 hours of training on a single NVIDIA 3090 Ti. We believe that the performance improvement over a stronger baseline model lends more credibility to our method.
>
> Furthermore, we observed a notable gap between Traj-MAE's validation and test performance on Argoverse 1. While its validation performance surpassed that of ProphNet and our SEPT, its test performance was significantly worse (as elaborated in our response to your Q3). In light of this, we find Traj-MAE's ablation study, which relies on their validation performance, less convincing.

---

> ### Author Response · Authors · 2023-11-14
> **Response to Reviewer fR6X (3/3)**
>
> > Q3: On the other hand, it seems that in Traj-MAE, its pretraining method improved the baseline's 0.732 minADE to a final 0.604 minADE, marking a relative decrease of 17.5%. Would this imply that Traj-MAE's pretraining technique holds a comparative advantage or superiority over SEPT's methodology?
>
> We did take note of Traj-MAE's performance on the Argoverse 1 validation set, where they achieved a remarkable minADE of 0.604. However, it's important to consider that Traj-MAE has not made their code publicly available, which makes it challenging to draw definitive conclusions about the differences between their pretraining methodology and SEPT's. We can offer some speculations based on our experience and observations:
> 1. In our experience, pretraining or other techniques typically lead to balanced improvements in both minADE and minFDE. For example, in the case of SEPT, we observed a 6.4% improvement in minADE and an 8.5% improvement in minFDE compared to the baseline without pretraining. In contrast, Traj-MAE achieved a 17.5% reduction in minADE but only an 8.5% reduction in minFDE, which appears unbalanced.
> 2. It's worth noting that, in our experience, even with some degree of distribution mismatch, improvements on the validation set tend to reflect a similar level of improvement on the test set. To provide a more comprehensive comparison, we analyzed the performance of SEPT, ProphNet, and Traj-MAE, as shown in the table below. ProphNet ranks 3rd on the Argoverse 1 leaderboard, and we observed that SEPT and ProphNet exhibit a similar pattern, while Traj-MAE's performance deviates significantly, particularly in terms of minADE.
>     |    |minADE (val)|minADE (test)|minADE gap|minFDE (val)|minFDE (test)|minFDE gap|
>     |:----|:----|:----|:----|:----|:----|:----|
>     |SEPT|0.643|0.728|11.7%|0.927|1.057|12.3%|
>     |ProphNet|0.680|0.762|10.8%|0.970|1.134|14.5%|
>     |Traj-MAE|0.600|0.810|**25.9%**|1.003|1.250|**19.8%**|
>
> Based on these two observations, we speculate that Traj-MAE may have encountered some form of data leakage during their pretraining process, such as accidentally incorporating validation set labels, which could have led to the observed improvement. Therefore, the substantial improvement in minADE might not necessarily prove the inherent advantage of their training scheme. It would be beneficial to have access to their code and further details to make a more accurate assessment.

---

> ### Comment · Reviewer_fR6X · 2023-11-21
>
> Thank you for your response. My concerns have been mostly resolved, and I will raise my score.

---

### Official Review · Reviewer_JZgu · 2023-11-04

**Soundness:** 4 excellent
**Presentation:** 4 excellent
**Contribution:** 3 good
**Rating:** 8
**Confidence:** 4

**Summary:**

This paper presents a modeling framework (SEPT) that leverages self-supervised learning to extract spatiotemporal relationships for traffic motion prediction. It involves three masking-reconstruction modeling tasks on scene inputs including agents’ trajectories and road network, pretraining the scene encoder to capture kinematics within a trajectory, spatial structure of road network, and interactions among roads and agents. The proposed approach is evaluated on two public datasets, i.e., Argoverse 1 and Argoverse 2, and achieves state-of-the-art performance.

**Strengths:**

- This work is well motivated in that the encoders built on universal architectures can develop a strong comprehension of traffic scenes through a properly designed training scheme. It leverages self-supervised learning to progressively develop the spatiotemporal understanding of traffic scenes.
- The proposed method is novel and technically sound.
- The effectiveness of the proposed SEPT is demonstrated by solid experiments on two of the largest datasets. It achieves significant improvement over the model learned from scratch on all motion forecasting metrics consistently.
- Extensive ablation study shows that the three tasks collaborate and yield positive effects on the final performance.
- This paper provides a good literature review, problem formulation, and discussion.
- This paper is well organized and presented, and thus easy to follow.

**Weaknesses:**

- When it comes to model efficiency, the paper only compares with QCNet which is one of the largest models among all comparison methods. Since time-consuming is one of the main drawbacks of previous methods as is claimed in Section 1, it would be good to know the exact inference speeds of SEPT and the other comparison methods.
- It would be good if the authors could talk more about their research findings in addition to presenting their approach, which I believe will inspire other researchers.

**Questions:**

N/A

---

> ### Author Response · Authors · 2023-11-14
> **Response to Reviewer JZgu**
>
> Thank you for your detailed assessment and positive evaluation of our paper. We carefully address each of your concerns as follows.
>
> > When it comes to model efficiency, the paper only compares with QCNet which is one of the largest models among all comparison methods. Since time-consuming is one of the main drawbacks of previous methods as is claimed in Section 1, it would be good to know the exact inference speeds of SEPT and the other comparison methods.
>
> Our paper primarily focuses on self-supervised scene understanding, which enhances prediction accuracy with a greatly simplified network design. We included QCNet in our comparisons because it was the previous SOTA on Argoverse 1 and 2 leaderboard, and intended to highlight that our method achieves SOTA performance with a reduced computational cost. In the paper, we indeed mentioned that “These empirical techniques consequently lead to an intricate and time-consuming information processing pipeline”. However, we acknowledge that (1) time-saving is not the primary objective of our research, and (2) the information processing pipeline is not the sole determinant of inference efficiency. Therefore, we plan to remove the claim of "time-consuming" from that sentence in our revised paper.
>
> In response to your inquiry about the exact inference speeds of SEPT, we conducted comprehensive tests under various circumstances, as outlined below:
> 1. GPU inference during training (batch size = 96): 26 iteration/s, or 2496 samples/s, on an NVIDIA RTX 3090 Ti;
> 2. GPU inference at deployment using ONNX (batch size = 1): 4.5 ms/sample, on an NVIDIA GTX 1060;
> 3. CPU inference at deployment using ONNX (batch size = 1, single-threaded): 27.8 ms/sample, on an Intel Core i5-8400.
>
> These results demonstrate that our model's inference speeds are competitive and suitable for use within an autonomous driving pipeline.
>
> > It would be good if the authors could talk more about their research findings in addition to presenting their approach, which I believe will inspire other researchers.
>
> We appreciate your feedback and agree with your suggestion. In our revised manuscript, we will expand our discussion to include a dedicated part on our research findings. This part will emphasize the significance of pretraining and its task design in the context of our work, addressing the concerns raised by multiple reviewers.

---

### Author Response · Authors · 2023-11-18
**Revision summary**

Thank you for your insightful reviews of our paper. We sincerely appreciate the constructive feedback, which guided our revisions aimed at enhancing the paper's quality. The following key changes have been implemented based on reviewers' valuable comments:
- Remove the "time-consuming" claim in the introduction section.
- Reference Traj-MAE and Forecast-MAE in the introduction section.
- Add references to MTR, HDGT and mmTransformer in the related works section.
- Expand the discussion of the relationship between our approach and Traj-MAE and Forecast-MAE in the related works section, to further elucidate the contributions of our proposed method.
- Add an ablation study to investigate the impact of the pretraining dataset in the appendix.

---

### Author Response · Authors · 2023-11-23
**We appreciate all reviewers' and AC's time and efforts**

We sincerely appreciate all reviewers' and AC's time and efforts in reviewing our paper. We truly thank you all for the insightful and constructive feedback, which helped to improve the quality of our paper.

Best wishes,

Authors

---

### Meta-Review · Area_Chair_i8NE · 2023-12-11

**Metareview:**

This submission leverages self-supervised learning for spatial-temporal modeling of traffic scenes.  Reviewers all appreciated the engineering efforts and results; meanwhile, they also pointed out that the technical innovations over prior work are limited.  The AC agrees with the reviewers' recommendations of acceptance.  The authors are encouraged to revise the submission based on the reviews for the camera-ready version.

**Justification For Why Not Higher Score:**

Reviewers pointed out that the technical innovations over prior work are limited.

**Justification For Why Not Lower Score:**

All reviewers recommend acceptance.

---

### Decision · Program_Chairs · 2024-01-16

Accept (poster)